# Successful Use of Geochemical Tools to Trace the Geographic Origin of Long-Snouted Seahorse *Hippocampus guttulatus* Raised in Captivity

**DOI:** 10.3390/ani11061534

**Published:** 2021-05-25

**Authors:** Ana Elisa Cabral, Fernando Ricardo, Carla Patinha, Eduardo Ferreira da Silva, Miguel Correia, Jorge Palma, Miquel Planas, Ricardo Calado

**Affiliations:** 1ECOMARE, CESAM-Centre for Environmental and Marine Studies, Department of Biology, University of Aveiro, Santiago University Campus, 3810-193 Aveiro, Portugal; anacabral@ua.pt (A.E.C.); fafr@ua.pt (F.R.); 2GEOBIOTEC—Department of Geosciences, University of Aveiro, Santiago University Campus, 3810-193 Aveiro, Portugal; cpatinha@ua.pt (C.P.); eafsilva@ua.pt (E.F.d.S.); 3CCMAR Centro de Ciências do Mar, Universidade do Algarve, Campus de Gambelas, 8005-139 Faro, Portugal; mtcorreia@ualg.pt (M.C.); jpalma@ualg.pt (J.P.); 4Department of Ecology and Marine Resources, IIM-CSIC-Instituto de Investigaciones Marinas (IIM), 36208 Vigo, Spain; mplanas@iim.csic.es

**Keywords:** bony plates, elemental fingerprints, ICP-MS, traceability

## Abstract

**Simple Summary:**

Seahorses (*Hippocampus* spp.) are currently exposed to a multitude of anthropogenic pressures worldwide. The illegal, unreported, and unregulated (IUU) fisheries and trade of these flagship species undermine the efforts to manage and protect their wild populations. Here we aim to validate a forensic tool to identify the geographic origin of seahorses and contribute to the ongoing fight against the illegal capture and trade of these organisms. The elemental fingerprints of long-snouted seahorse (*Hippocampus guttulatus*) bony structures, including the subdermal bony plates that cover their body, revealed that they can be successfully employed to confirm their geographic origin. The results of this first study using seahorses raised in captivity indicate that this tool may also allow to discriminate between different populations of wild specimens and enhance the traceability of traded specimens.

**Abstract:**

The global market of dried seahorses mainly supplies Traditional Chinese Medicine and still relies on blurry trade chains that often cover less sustainable practices targeting these pricey and endangered fish. As such, reliable tools that allow the enforcement of traceability, namely to confirm the geographic origin of traded seahorses, are urgently needed. The present study evaluated the use of elemental fingerprints (EF) in the bony structures of long-snouted seahorses *Hippocampus guttulatus* raised in captivity in two different locations (southern Portugal and Northern Spain) to discriminate their geographic origin. The EF of different body parts of *H. guttulatus* were also evaluated as potential proxies for the EF of the whole body, in order to allow the analysis of damaged specimens and avoid the use of whole specimens for analysis. The contrasting EF of *H. guttulatus* raised in the two locations allowed their reliable discrimination. Although no single body part exactly mimicked the EF of the whole body, seahorse trunks, as well as damaged specimens, could still be correctly allocated to their geographic origin. This promising forensic approach to discriminate the geographic origin of seahorses raised in captivity should now be validated for wild conspecifics originating from different locations, as well as for other species within genus *Hippocampus*.

## 1. Introduction

Seahorses (*Hippocampus* spp.) are marine teleost fishes that along with seadragons, pipehorses, and pipefishes, form the family Syngnathidae [1]. These bony fishes have an unusual phenotype, featuring horse-shaped heads, a curved trunk, no caudal fin, and a prehensile tail that allows them to grasp holdfast [2,3]. Seahorses’ bodies are covered by subdermal bony plates, which allow for axial bending and the prehensility of the tails, as well as protection of their spinal column from compressive forces [3,4]. These structures are composed by calcium phosphate, with organic and water fractions [4]. The fish skeletal system is formed by bones and cartilage; vertebrae are mainly composed of calcium, phosphate and carbonate and, to a smaller extent, magnesium, sodium, strontium, lead, citrate, fluoride, hydroxide, and sulfate [5].

Seahorses are flagship species and important ambassadors of marine conservation [6]. These organisms are, however, susceptible to multiple anthropogenic pressures, such as habitat destruction and untargeted fisheries [6]. Besides inhabiting some of the most impacted marine coastal ecosystems (e.g., seagrass beds, coral reefs) due to human occupation, seahorses display a number of biological and ecological features that make them especially vulnerable to habitat loss and over-exploitation [2,7]. Some examples include heterogeneous distribution and habitat fidelity, poor swimming skills, monogamy, and low fecundity [7].

While seahorses are traded live to supply the marine aquarium industry, they can also be marketed dry for the curio trade; nonetheless, the overwhelming majority of seahorses are traded dry to supply demand by Traditional Chinese Medicine [8]. Managing seahorse fisheries and trade in a sustainable manner has proven to be a difficult task over the past decades, with several seahorse species still being termed as Data Deficient [9]. Tracing the origin of seahorses being traded to supply the demand for these pricey marine organisms is also a challenging task, although all seahorse species are currently included in the IUCN Red List of Threatened Species and in the Appendix II of CITES, the latter regulating their international trade [10,11]. Tracing the place of origin of live seahorses being traded to supply the marine aquarium industry is often difficult to achieve, as it is for most marine ornamentals [12]. While the preliminary use of bacterial fingerprints of live seahorse skin mucus may contribute to enhance the traceability of seahorses being traded live [13], this tool will likely be of little use to trace the geographic origin of specimens being traded dry for the curio trade and Traditional Chinese Medicine. For seahorses being traded dry the use of elemental fingerprints (EF) of their bony structures emerges as a more appropriate tool to trace their place of origin. Indeed, EF have already been successfully used to trace the geographic origin of seafood [14]. Additionally, these geochemical tools are already well-established to identify the place of origin of bony fish through the EF of their otoliths (ear bones) [15,16], as well as their scales and spines [17,18]. To the authors best knowledge, the use of EF has never been investigated as an approach to trace the place of origin of seahorses.

The present study aims to evaluate: (i) If the EF of bony structures in seahorses body parts (namely the head, trunk, dorsal fin or tail) can be used as a proxy for the EF of the whole body; (ii) if the EF of bony structures in a body part of seahorses cultured in different locations can be used to reliably trace their geographic origin; and (iii) if the EF of bony structures from seahorse portions (mimicking damaged specimens missing one or two body parts, as traded/apprehended organisms may not be whole) still allow to reliably trace their place of origin. It is expected that the use of EF may allow researchers to confirm or refute the geographic origin of seahorses, ultimately contributing to the ongoing fight against the illegal, unreported, and unregulated (IUU) fisheries and trade of these flagship species. The present approach is to be used under a forensics framework, when no information on the location of origin of seahorses and no seawater samples are made available. The question being addressed in the present study is, are EF of seahorses originating from different locations contrasting to the point of allowing their discrimination.

## 2. Materials and Methods

### 2.1. Biological Material

A total of 15 long-snout seahorse specimens (*Hippocampus guttulatus*) cultured in captivity were sourced from the Instituto de Investigaciones Marinas (IIM) (CSIC, Vigo, Spain) (*n* = 10) and from the Centro de Ciências do Mar (CCMAR) (University of the Algarve, Portugal) (*n* = 5). The specimens from IIM were cultured in natural seawater from Ría de Vigo located on the Atlantic margin of south western Galicia (Northern Spain) (42°21 N; 8°36′ and 8°54′ W), while the specimens from CCMAR were cultured in natural seawater from Ria Formosa coastal lagoon, located in the south of Portugal (36°59′ N; 7°51′ W).

Based on the body parts described by Lourie [3], five of the ten specimens from IIM were subdivided into the following four body parts: head, trunk, dorsal fin, and tail (4 body parts × 5 specimens = 20 samples) (Figure 1). The remaining five specimens from IIM were kept whole to test if elemental fingerprints (EF) of a seahorse body part could be used as a proxy of the whole body (5 whole seahorses = 5 samples). The EF in seahorse bony structures was compared between specimens cultured at IIM (Spain) and CCMAR. For this purpose, five specimens originating from each research center were divided into the four body parts as described above (2 locations × 4 body parts × 5 replicates = 40 samples) (specimens from IIM used for this comparison were the same used for comparing against the EF of whole seahorses described above). All specimens were provided frozen (died from natural causes in the laboratory) and were maintained at −20 °C until further analysis.

### 2.2. Samples Preparation

The organic components (e.g., soft tissues and skin) of seahorse body parts were removed using ceramic coated blades and tweezers, in order to prevent metal contamination [19]. Subsequently, samples were freeze-dried (≈24 h) and bones from the head, trunk, tail, and the whole body were individually homogenized using a mortar grinder (RM 200, Retsch, Hann, Germany). The dorsal fin was manually homogenized using a ceramic mortar and pestle. The mortar grinder and the mortar and pestle were cleaned between samples with quartz powder followed by alcohol (70%), to avoid potential cross-contamination [19]. For posterior digestion of samples, a subsample weighting proximately 0.045 g was used for the whole body, head, trunk, and tail, while the whole dorsal fin was employed.

### 2.3. Elements Extraction and ICP-MS Analysis

The digestion of samples was performed by adding 2 mL of concentrate nitric acid (PanReac Applichem, Chicago, IL, USA) purified by sub-boiling distillation (Savillex-DST-1000, Eden Prairie, MN, USA), 0.75 mL of hydrochloric acid (Fisher Chemical, Hampton, New Hampshire, NH, USA) and 2 mL of hydrogen peroxide (Labem, Spain) (3:1:3 *v/v/v*). Following acidic digestion (overnight 14–16 h), solutions were heated on a digestion block (DigiPrep, SCP Science; Montreal, QC, Canada). These solutions were initially exposed, for 10 min, to an increase from room temperature to 50 °C that was kept stable for 15 min. Temperature was then increased from 50 °C to 85 °C during a period of 15 min. When temperature stabilized at 85 °C, it was maintained for 15 min, until the end of the cycle. After the first digestion, 2 mL of H_2_O_2_ were added to remove traces of organic matter and a second heating cycle, with the same characteristics described above, was applied. The drying process of the samples was performed subsequently, at a temperature of 45 °C, in the digestion block. Following evaporation and to avoid calcium (Ca) from masking the concentration of other elements [20], samples were diluted with 25 mL of Milli–Q (Millipore) water to a final acid concentration of 1% HNO_3_. Moreover, a dilution consisting in the addition of 3 mL of HNO_3_ at 1% to 1.5 mL of the samples’ solution, was prepared and placed in plastic vial tubes for elemental reading in the inductively coupled plasma mass spectrometry (ICP-MS) equipment. 

The total concentration of 15 elements [Al (aluminum), Ba (barium), Ca (calcium), Ce (cerium), Cr (chromium), Cu (copper), Fe (iron), K (potassium), Mg (magnesium), Mn (manganese), Na (sodium), Ni (nickel), P (phosphorus), Sr (strontium), and Zn (zinc)] (Appendix A), was determined using ICP-MS, in an Agilent 7700 ICP-MS (Agilent Technologies, Santa Clara, CA, USA) equipped with an octopole collision cell and autosampler. Elements were quantified through calibrations with standard solutions of each analyte (High-Purity Standards—ISC Science, Oviedo, Spain). The analysis performed were validated through a quality control program for the determination of each element being surveyed, including method blanks, and replicate samples. Accuracy of the ICP-MS method was evaluated by the analysis of a certified reference material BCS-CRM-513 (SGT Limestone 1). Recovery of elements was acceptable: Mg (116%), Ca (120%), Mn (118%), Fe (93%), Sr (119%), Ba (103%). Precision was estimated by the relative standard deviation (RSD) of five replicate samples, being ≤10%.

### 2.4. Statistical Analysis

The concentration values of all elements recorded in the bony structures of *H. guttulatus* were standardized to Ca (mmol/mol Ca) and the statistical analysis was carried out on element ratios (X: Ca) [21]. Six seahorse body portions, mimicking damaged specimens (1—head + trunk, 2—head + trunk + dorsal fin, 3—head + trunk + tail, 4—trunk + dorsal fin, 5—trunk + dorsal fin + tail, and 6—trunk + tail), as well as the equivalent to the whole body (head + trunk + dorsal fin + tail), were obtained by adding the elemental concentrations determined on the four original body parts, from specimens originating from the two locations (IIM in Galicia, Spain and CCMAR in the Algarve, Portugal). All combinations of body portions were performed using portions from the same individual seahorse, thus allowing to always maintain individual specimens identifiable throughout the statistical analysis. The resemblance matrix was calculated based on the normalized Euclidean distance among samples from IMM and CCMAR. To assess if the head, trunk, dorsal fin or tail were a reliable proxy of seahorses’ whole body, a one-way permutational analysis of variance (PERMANOVA) test [20] was performed, using body parts as the single factor (*n* = 5). A two-way crossed PERMANOVA was performed to evaluate the existence of significant differences in the EF of seahorses using body parts as fixed factor with four levels (head, trunk, dorsal fin, and tail) and sampling locations also as fixed factor with two levels (IMM and CCMAR) (*n* = 5). This same analysis was also performed to detect the significant differences in the EF of seahorse body portions (specimens missing one or two body parts), belonging to seahorses from each of the two locations. The PERMANOVA tests were run with 9999 Monte Carlo permutations. The pseudo-F values in the main tests and the *t*-statistic in the pairwise comparisons were evaluated in terms of the significance among levels of the tested factor. Values of *p* ≤ 0.05 reveal that the groups differ significantly. Similarity percentages (SIMPER) [20] were also analyzed to describe the level of similarity of EF of whole seahorse specimens and their body parts within the same location and between the two different locations (*n* = 5). A PCO analysis [20] was performed to visualize the inter-individual spatial differences in EF among the five groups of samples (head, trunk, dorsal fin, tail, and whole body) from IIM (*n* = 5). This method was also applied to assess the existence of inter-individual spatial differences in EF among the groups of the four body parts, from the two locations (*n* = 5). Random forest analysis was performed to assess the reliability of using EF from seahorse bony structures to infer their geographic origin. For this purpose, equivalents to the whole body (head + trunk + dorsal fin + tail), were marked with their respective origins (IIM in Galicia, Spain and CCMAR in the Algarve, Portugal), while the body parts and body portions, from both locations, were considered to be blind samples. A random forest prediction model using samples from the two locations, (5 replicates per location = 10 samples) was built and samples of the whole body from IIM, considered as blind samples, were evaluated by classification on this model.

PERMANOVA, PCO and SIMPER were performed using PRIMER v7 (Primer-e, Auckland, New Zealand) with the add-on PERMANOVA+, while random forest analysis were performed using R version 3.6.1 (Vienna, Austria).

## 3. Results

### 3.1. Assessment of a Proxy to the Whole Body

The one-way PERMANOVA analysis between the four body parts (head, trunk, dorsal fin, and tail) and the whole body of seahorses cultured in IIM revealed the existence of significant differences (Main test: Pseudo-F = 12.900, *p* = 0.001), with pair wise comparisons indicating that the EF of body parts differed significantly from that displayed by the whole body of seahorses (Table 1).

The first two axes of the PCO analysis explained 85.5% of EF variation in the data set (PCO axis 1: 76.4%, PCO axis 2: 9.2%) (Figure 2). The dorsal fin appeared as a distinctive group, as shown in Figure 2.

### 3.2. Traceability of the Geographic Origin

#### 3.2.1. Elemental Composition of Seahorses from IIM and CCMAR

The distribution of the maximum and minimum concentrations of the 14 elements to Ca present in the samples analyzed varied between specimens from IIM and CCMAR (Appendix A). The highest ratios of Al/Ca, Ba/Ca, Ce/Ca, Cr/Ca, Cu/Ca, Fe/Ca, and Mn/Ca were recorded in the dorsal fin of specimens from both locations. On the other side, specimens from IIM and CCMAR exhibited the lowest concentrations of Cr/Ca, Cu/Ca, Fe/Ca in the trunk and of Sr/Ca in their dorsal fin. The two-way crossed PERMANOVA revealed the existence of significant differences between the EF of the bony structures of the same body parts (head, trunk, dorsal fin, or tail) of seahorses from IIM and CCMAR, with a significant interaction between the two fixed factors (body parts and sampling locations) (Main test: Pseudo-F = 8.786, *p* = 0.001). Pair-wise comparisons indicated that only the EF of heads of seahorses from the two locations did not differ significantly (Table 2).

The first two axes of the PCO analysis explained 75.2% of the EF variation of the four body parts, from IIM and CCMAR (PCO axis 1: 65.4%, PCO axis 2: 9.8%) (Figure 3). The dorsal fins of seahorses from the two locations appeared as distinctive groups, as shown in Figure 3.

Results for the SIMPER analysis comparing the EF of different seahorse body parts are summarized in Appendix A. It is worth highlighting that the most contrasting EF was displayed by seahorse dorsal fins from the two geographic origins, although closely followed by seahorse trunks. The chemical elements contributing the most for the contrasting EF recorded in seahorse dorsal fins were (in decreasing order of relevance) Al, Ni, Na, Ba, and Mn that accounted for nearly 60% of the dissimilarities recorded. Concerning the differences recorded in seahorse trunks, these were mostly due to P (62.67%) and Sr (16.34%).

#### 3.2.2. Body Parts as Blind Samples

The random forest analysis, with only the equivalent to the whole bodies marked as known origin, resulted in the correct allocation of the place of origin of all trunk replicates from seahorses originating from the two locations (Table 3). All dorsal fins from seahorses cultured at IIM were erroneously assigned to CCMAR (Table 3). Concerning specimens raised at CCMAR, part of the heads and tails screened were erroneously assigned to IIM (Table 3).

#### 3.2.3. Validation of the Trunk Elemental Fingerprint as a Model to Trace the Geographic Origin of Seahorses

The random forest analysis that included all trunk replicates from both locations (IIM and CCMAR) and the organisms processed whole from IIM, used as blind samples, yielded a success classification rate of 100%, with all whole-body replicates being correctly allocated to IIM.

#### 3.2.4. Elemental Composition of Seahorse Body Portions

The two-way crossed PERMANOVA revealed the existence of significant differences between the EF of the same seahorse body portions (seahorses missing one or two body parts mimicking damaged specimens) originating from IIM and CCMAR, with a significant interaction between the two fixed factors (body portions and sampling locations) (Main test: Pseudo-F = 84.549, *p* = 0.001). Pair-wise comparisons indicated that the EF of body portions from the two locations differed significantly between them (Table 4).

#### 3.2.5. Seahorse Body Portions as Blind Samples

The random forest analysis, with only the equivalents to the whole body marked as known origin yielded a correct allocation of all replicates of the six seahorse body portions considered (1—head + trunk, 2—head + trunk + dorsal fin, 3—head + trunk + tail, 4—trunk + dorsal fin, 5—trunk + dorsal fin + tail, 6—trunk + tail) to IIM and CCMAR.

## 4. Discussion

The combat to seahorse illegal trade remains a challenging task, due to the number of parties involved and the submission of incomplete or ambiguous reports to CITES [11]. The analysis of EF may allow to combat the illegal and fraudulent trade of seahorses by allowing to trace and confirm their geographic origin, with a relatively faster and less expensive approach when compared to other methods employed (e.g., biochemical and molecular tools) [14]. Furthermore, given the importance of working in sustainable alternatives to the collection of seahorses from the wild (e.g., commercial aquaculture) and the certification of cultured specimens, the use of the geochemical composition of seahorse bony structures may be an important tool to discriminate wild specimens from cultured ones. In fact, a study developed by Arechavala-Lopez et al. [22], in which otolith trace elemental compositions from cultured and wild seabass (*Dicentrarchus labrax*) and gilthead sea bream (*Sparus aurata*) were compared, revealed that elemental fingerprinting can be used to successfully discriminate farmed fish from wild conspecifics.

Previous studies addressing seahorse bony structure have mainly focused on its development and ossification processes [23], the morphological variation in seahorse vertebral system [24,25], and deformation mechanisms of its bony armor [4,26]. The use of the geochemical composition of seahorse bony structures to trace their geographic origin has never been previously explored. However, the use of geochemical tools to trace the geographic origin and habitat use, has been employed for endangered species of fish [27] and other animal groups [28,29].

The dorsal fin of specimens from both locations was the body part that presented the higher values of Al/Ca, Ba/Ca, Ce/Ca, Cr/Ca, Cu/Ca, Fe/Ca, and Mn/Ca (Appendix A). According to Avigliano et al. [29], ratios of metal/Ca, such as Cu/Ca, Fe/Ca, Na/Ca, and Zn/Ca, of the edges of dorsal fin spines from the White Sea catfish (*Genidens barbus*) were considered to be potentially useful for the discrimination of habitat use by this species, as well as to discriminate its stocks. Additionally, natural element/Ca, as Ba/Ca, Mg/Ca, Mn/Ca, and Sr/Ca, present in the edges of dorsal fin spines of *G. barbus*, were also reported to be effective in stock identification [27]. The edges of fin spines present low rates of resorption, thus being able to successfully retain chemical elements fingerprinted throughout development [30]. Thus, whenever present in the environment, metals from anthropogenic origin, such as Cu, Pb, and Zn, may contribute to successfully allocate a given specimen to a more contaminated ecosystem [30]. As such, the potential use of the EF of seahorse dorsal fin is certainly worth investigating to trace their geographic origin, namely when specimens originate from areas with strong anthropogenic activities, such as estuaries and coastal lagoons. However, the dorsal fin was also the body part exhibiting the most contrasting EF in seahorses (see below), a feature that may either confirm or refute its potential use for the traceability of geographic origin. Given the preliminary nature of the present work, for now, one can only advocate that future works should further investigate this issue.

The one-way PERMANOVA confirmed that none of the tested body parts (head, trunk, dorsal fin, and tail) were a perfect proxy for the EF of the whole body of seahorses (Table 1). According to Kerr and Campana [31], the incorporation of some elements in fish vertebrae can occur through absorption across the skin and gills or through diet, whose contribution to the chemical composition of fish hard parts varies with element and type of structure. Nevertheless, the relative contribution of water and diet, as well as the proportion of elemental incorporation into the vertebrae, are not extensively studied [31]. The ratios Mg/Ca and Mn/Ca, were the ones that least varied between the whole body and the body parts, with Mg being one of the main components of fish vertebrae [5]. The PCO analysis clearly highlighted that the dorsal fin was the body part that displayed the highest variability of its EF and whose EF was less similar to all other seahorse body parts compared.

While the EF of no seahorse body part studied can be considered an optimal proxy of the EF displayed by their whole body, the two-way crossed PERMANOVA evidenced that with the exception of heads, the EF of the same body parts of seahorses from IIM and CCMAR were significantly different. These results support the possibility of using some body parts of a seahorse to trace its geographic origin. Indeed, this finding sets the starting point to pursue the traceability of seahorses’ geographic origin using the EF of their bony structures. 

The random forest analysis revealed the correct allocation of all seahorse trunks to their respective place of origin (Table 3), a finding in line with the low level of similarity revealed by the SIMPER analysis when comparing the EF of these body parts sampled from seahorses from IIM and CCMAR. Although dorsal fins of seahorses originating from CCMAR were all correctly classified, this was not the case for the same body parts of specimens originating from IIM. In fact, none of the dorsal fins from seahorses cultured at IIM were correctly assigned to their place of origin (Table 3). As already mentioned above, the higher level of variability recorded in the EF of seahorses’ dorsal fin, may either be an advantage or a constraint to advocate its use for the traceability of their place of origin. This issue may likely only be clarified after further studies addressing this topic are performed. Overall, despite the EF of no body part being a perfect proxy of EF of the whole body of a seahorse, the present study suggests that the body part that holds the highest potential to successfully allocate a given seahorse specimen to its correct place of origin is most likely the trunk.

Several factors may affect the chemical composition of calcified structures of bony fish, such as water chemistry [31], diet [31], environmental conditions (e.g., temperature) [32], physiology [31], fish size [31], and even genetic aspects [33]. Additionally, the bone is not a metabolically inert structure, as elemental mobilization from the bone tissue is known to occur [34]. Elemental fingerprints can also vary according to habitat use (e.g., habitat use may differ with life cycle stage) [35]. Coastal ecosystems, such as estuaries with lower oceanic influences, present a more variable physical and chemical environment, when compared with the more homogenous nature of the deep ocean [32]. This factor can be reflected in the geochemical fingerprints displayed by populations of marine organisms, since coastal ecosystems are often expressed in more variable EF present in the mineralized structures [32,36]. Seahorses usually inhabit shallow coastal waters [37], which can promote distinct EF between populations. Water chemistry can vary depending on the spatial and temporal scales (e.g., seasonal and interannual), a feature known to be often displayed in otolith chemistry [31]. Furthermore, temporal variability of water chemistry can differently shape the EF of individuals from the same population, due to intrinsic traits, such as size [31]. However, in the present study, although the water supplied was from natural sources, the chemical and physical conditions were maintained stable during culture in the laboratory, thus reducing, or even eliminating, variability promoted by temporal shifts. The fact that seahorses reared under controlled conditions in two different laboratories could be successfully discriminated through the EF of their bony structures opens good perspectives for the successful use of this geochemical tool for seahorse traceability. The successful use of this approach was also confirmed by the random forest analysis performed using different body portions of seahorses (mimicking damaged specimens), which further confirmed the potential of the EF of their bony structures to trace the geographic origin of seahorses.

## 5. Conclusions

The results of this preliminary research suggest that the EF of the bony structure of seahorses can be successfully used to trace their geographic origin. The use of seahorse trunks may potentially be an alternative to using the whole body, with its processing per se being an advantage in terms of time and cost to prepare and process samples. Furthermore, the edge of the dorsal fin may be confirmed in future studies as a suitable matrix to trace seahorse populations known to occur in more polluted environments, with fin clipping already being a well-established method that does not promote significant deleterious effects on the growth and survival of fish species, including seahorses [38]. In case of apprehension, the EF of damaged seahorses may still be used to successfully trace their geographic origin. Nevertheless, it is important to validate the use of EF of seahorses’ bony structures as a tool for traceability of their place of origin using wild specimens, as the present study solely surveyed cultured seahorses. As wild conspecifics are known to be exposed to more dynamic environments (from a biotic and abiotic perspective), it is important to determine if the current approach remains valid. Future studies should aim to employ a larger number of seahorse specimens from different geographic origins, both from the wild and different aquaculture laboratories/companies. When using a larger number of specimens to validate the approach here presented, sophisticated feature selection methods (wrapper methods or embedded selection methods) can be used to eliminate the less relevant elemental ratios to Ca from the analysis and help pinpointing the most important ones for discrimination of geographic origin. If not suitably selected, large numbers of predictive variables may at times have a confounding effect, which may ultimately impair the discrimination of different geographic origins. Moreover, additional seahorse species should be screened, with emphasis on the comparison of species occurring in sympatry, such as *H. gutullatus* and *H. hippocampus*. The potential effect of intrinsic traits, such as size and gender, on the EF of seahorse specimens being studied should also be addressed in future works. The existence of seasonal and interannual variability in EF must also be investigated, as this will be paramount to establishing a potential baseline for geographic origins considered as a priority for seahorse conservation (e.g., marine protected areas). If such a baseline can be established, the verification of claims on geographic origin, as well as the identification of areas more prone to pouching, could be more easily performed using EF. The spatial resolution that may be achieved when using the approach here described is also another topic worth investigating (will it be possible to discriminate between habitats within a same ecosystem?). It is also relevant that researchers start to unravel the drivers and mechanisms that shape the EF of the bonny structures of seahorses to allow answering the question: why are EF different? 

## Figures and Tables

**Figure 1 animals-11-01534-f001:**
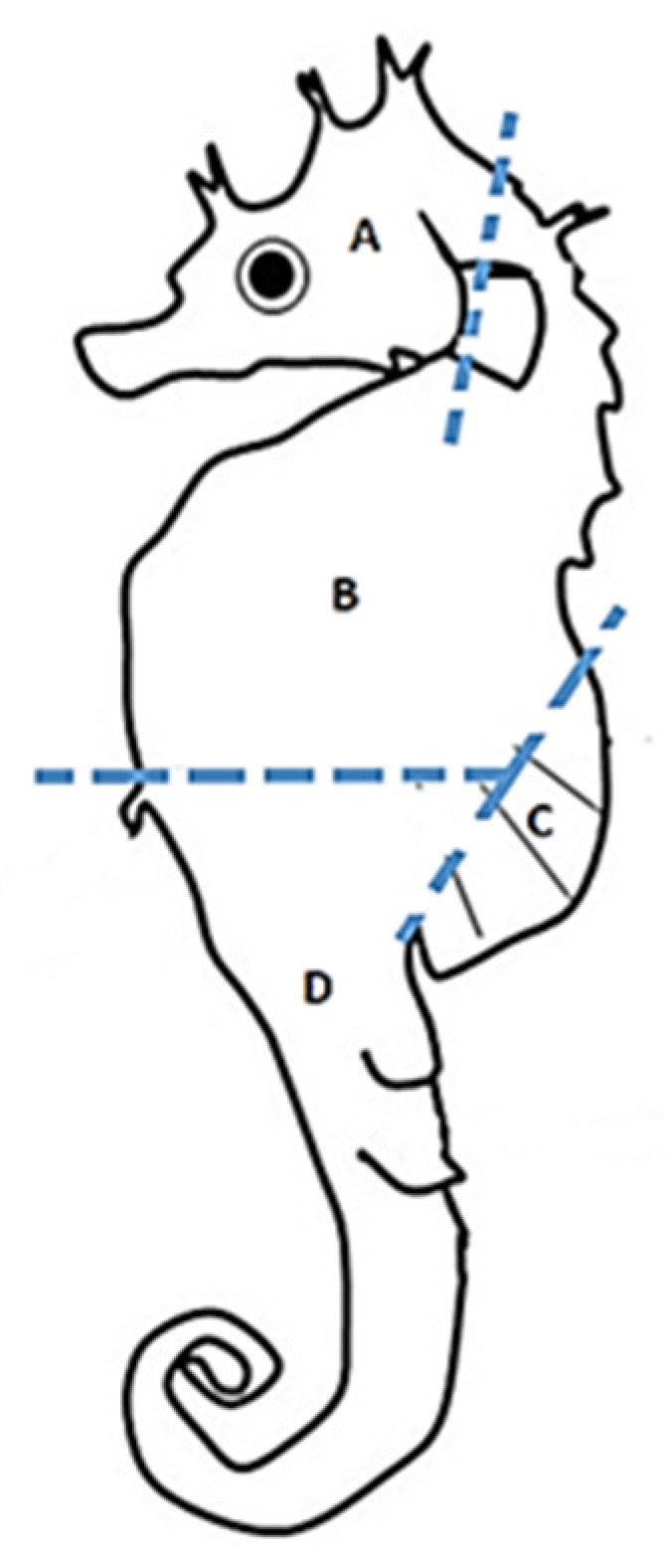
Schematic representation of the four body parts analyzed for long-snouted seahorse (*Hippocampus guttulatus*). **A**—head, **B**—trunk, **C**—dorsal fin and **D**—tail.

**Figure 2 animals-11-01534-f002:**
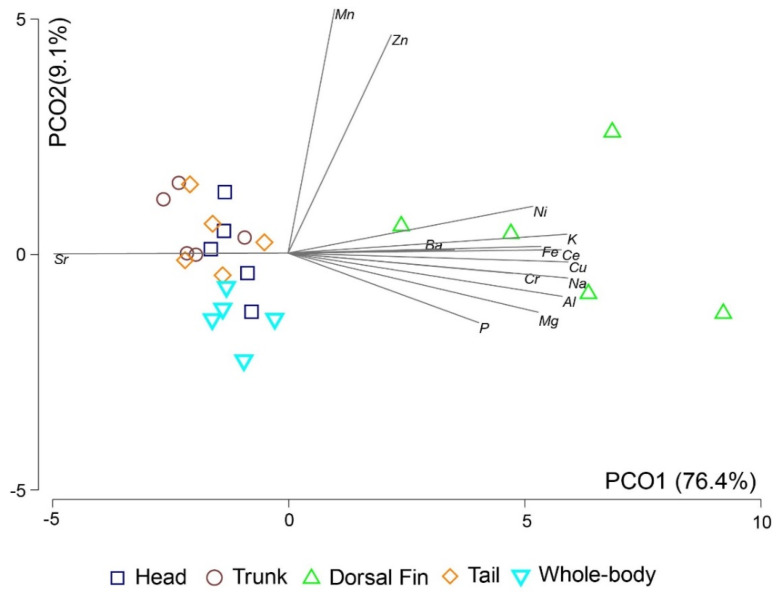
Principal coordinates analysis of the elemental fingerprints of the bony structures of four body parts (head, trunk, dorsal fin, and tail) and the whole body of long-snouted seahorses (*Hippocampus guttulatus*) cultured in Instituto de Investigaciones Marinas (IIM) (Galicia, Spain) (*n* = 5). Vector overlay Pearson correlations of elemental composition with PCO axes are shown if |r| > 0.20.

**Figure 3 animals-11-01534-f003:**
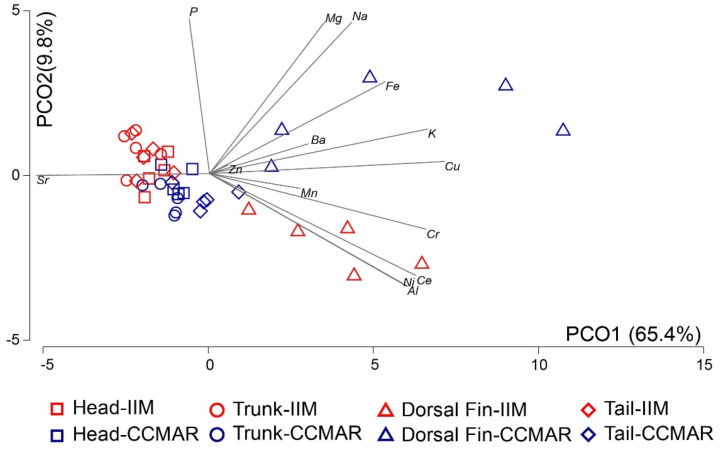
Principal coordinates analysis of the elemental fingerprints of four body parts (head, trunk, dorsal fin, and tail) of long-snouted seahorses (*Hippocampus guttulatus*) from Instituto de Investigaciones Marinas (IIM) (Galicia, Spain) and Centro de Ciências do Mar (CCMAR) (Algarve, Portugal) (*n* = 5). Vector overlay Pearson correlations of elemental composition with PCO axes are shown if |r| > 0.20.

**Table 1 animals-11-01534-t001:** Pair-wise comparison of permutational multivariate analysis of variance (PERMANOVA) between the elemental fingerprints of the bony structures of four body parts (head, trunk, dorsal fin, and tail) and the whole body of long-snouted seahorses (*Hippocampus guttulatus*) cultured in Instituto de Investigaciones Marinas (IIM) (Galicia, Spain). (*n* = 5).

Pair-Wise Comparisons	Permanova	
	*t*	*p*
Head vs. Whole-body	1.710	0.046
Trunk vs. Whole-body	2.486	0.004
Dorsal fin vs. Whole-body	4.163	0.001
Tail vs. Whole-body	2.030	0.013

**Table 2 animals-11-01534-t002:** Pairwise comparisons of two-way crossed permutational multivariate analysis of variance (PERMANOVA) between the elemental fingerprints of the bony structures of four body parts (head, trunk, dorsal fin and tail) of long-snouted seahorses (*Hippocampus guttulatus*) from Instituto de Investigaciones Marinas (IIM) (Galicia, Spain) and Centro de Ciências do Mar (CCMAR) (Algarve, Portugal) (*n* = 5).

Pair-Wise Comparisons	Permanova	
	*t*	*p*
Head IIM vs. Head CCMAR	1.440	0.130
Trunk IIM vs. Trunk CCMAR	3.565	<0.001
Dorsal Fin IIM vs. Dorsal Fin CCMAR	1.863	0.039
Tail IIM vs. Tail CCMAR	2.656	0.002

**Table 3 animals-11-01534-t003:** Classification success by origin of sample (IIM or CCMAR) of a random forest analysis based on the elemental fingerprints of the four body parts (head, trunk, dorsal fin, and tail) of long-snouted seahorses (*Hippocampus guttulatus*) from Instituto de Investigaciones Marinas (IIM) (Galicia, Spain) and Centro de Ciências do Mar (CCMAR) (Algarve, Portugal) (*n* = 5).

Origin of Sample	IIM (%)	CCMAR (%)
IIM-Head	100	0
IIM-Trunk	100	0
IIM-Dorsal Fin	0	100
IIM-Tail	100	0
CCMAR-Head	60	40
CCMAR-Trunk	0	100
CCMAR-Dorsal Fin	0	100
CCMAR-Tail	40	60

**Table 4 animals-11-01534-t004:** Pair-wise comparisons of two-way crossed permutational multivariate analysis of variance (PERMANOVA) between the elemental fingerprints of the same body portions (seahorses missing one or two body parts mimicking damaged specimens) of long-snouted seahorses (*Hippocampus guttulatus*) from Instituto de Investigaciones Marinas (IIM) (Galicia, Spain) and Centro de Ciências do Mar (CCMAR) (Algarve, Portugal). H-head, T-trunk, DF-dorsal fin, and TL-tail (*n* = 5).

Pair-Wise Comparions	Permanova	
	*t*	*p*
H + T + DF IIM vs. H + T + DF CCMAR	2.507	0.003
T + DF IIM vs. T + DF CCMAR	2.488	0.004
T + TL IIM vs. T + TL CCMAR	2.664	0.003
T + DF + TL IIM vs. T + DF + TL CCMAR	2.694	0.002
H + T IIM vs. H + T CCMAR	2.368	0.004
H + T + TL IIM vs. H + T + TL CCMAR	2.400	0.006

## Data Availability

All raw data of ICP-MS analysis is available as Appendix A (Appendix A).

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
