# Peer review of "Successful Use of Geochemical Tools to Trace the Geographic Origin of Long-Snouted Seahorse Hippocampus guttulatus Raised in Captivity"

_animals, 2021, doi:10.3390/ani11061534_

Round 1

Reviewer 1 Report

Line 95: Remove " in the beginning: The present approach is to be used under a forensics framework, when no information on the location of origin of seahorses and no seawater samples are made available. The question being addressed in the present study is “are EF of seahorses originating from different locations contrasting to the point of allowing their discrimination”. 

Line 99: remove last part of sentence

Author Response

Reviewer 1

Line 95: Remove " in the beginning: The present approach is to be used under a forensics framework, when no information on the location of origin of seahorses and no seawater samples are made available. The question being addressed in the present study is “are EF of seahorses originating from different locations contrasting to the point of allowing their discrimination”. 

Corrected as suggested.

Line 99: remove last part of sentence

Corrected as suggested.

Reviewer 2 Report

The authors have carried out a long-time needed study to aide in the identification of the geographic origin of seahorses and/or their body parts. The manuscript provides a well-supported justification for its clearly stated objectives. The introduction is concise and complete and they have gone so far as to explicitly specify the questions that the study intends to answer. Results are discussed in a complete and organised way, the concluding remarks are distinct and in accordance with their main findings, and further topics to continue this investigation are properly recognised. However, I do have a few important concerns regarding the statistical analysis on the data and the way in which results are presented and interpretations obtained. Whilst the multivariate approach to analyse the data is perfectly justified, there are several points that in my opinion need to be clarified before the ms can be accepted.

  • It is not clear why the authors applied univariate ANOVAs on the ratios of the elements, following the ANOSIM tests. The major advantage of a multivariate approach is the fact that it examines all variables (or descriptors) measured on n sampling units and their correlations in the same multidimensional space. This is, the multivariate procedure takes into account changes on all multidimensions at the same time. This, directly conflicts with univariate analyses, which take each variable in a single 1-dimensional space (i. e. as if all other variables did not exist) and are therefore bound to lose relevant information on the correlation between EF descriptors (which, in this case, are correlated because they were measured the same individuals). Results of ANOVAs are therefore not only redundant, but conflicting in their interpretation with the results of ANOSIMs and other multivariate procedures used (e.g. the multidimensional scaling used to visualize the data). (please refer to Chapter 1 in the Methods manual of PRIMER). If there is need to asses the relative contribution of EFs on the ability to distinguish the different groups, the authors should refer to methods such as SIMPER (used only in the case of section “3.2.1. Elemental composition of seahorse body portions”), or a close examination of the eigenvectors in the PCOs they have already applied.
  • It is not clear why authors apply a series of sequential ANOSIM tests (Tables 1, 2 and 4) and interpreted their results as if all tests were truly independent (i.e. p values were not adjusted to correct for increased probability of type I error with a Bonferroni correction, for example). This procedure is not only unjustified, but its interpretation as a whole is unreliable from a probabilistic perspective. Moreover, hypotheses concerning differences between more than two groups are perfectly testable using ANOSIM, followed by pairwise tests between groups, especially if a strategy that reduces tests to a minimum is kept (please refer to Chapter 6 in the Methods manual of PRIMER).
  • PCOs are lacking the vectors that represent the descriptors, a visual aide that would be very informative on the partial contribution of each descriptor to the ordination map (i. e. which EFs contribute most to separate samples in the first and second principal coordinates) (please see Chapter 3 in the Methods manual of PERMANOVA)
  • The ordination of samples obtained with the PCO (Figure 2) do not match the results of the sequential ANOSIMs (Table 1). Results of the latter suggest all groups are different (i.e. separated) from the “whole body”; however, the maps show marked overlap of all groups except “dorsal fin”. Also, it would be helpful to have a PCO with the results of the elemental composition of seahorse body portions.
  • It is difficult to identify the different experimental hypotheses that are being tested (differences between locations, between body parts, between body fractions and locations, etc.), with each one of the different tests applied, and the particular fraction of data set used for such purposes. As a result, it is not clear whether the underlying design for ANOSIM tests is a one-way (1 factor), a 2-factor nested (1 factor nested in another), a 2-way crossed (2 factors combined with or without replication). The differences amongst these have important implications in terms of how the test are constructed and the permutations performed (please see Chapter 6 in the Methods manual of PRIMER).
  • There are some apparent contradictions in the number of replicates used for certain tests. For example, there is ANOSIM and a PCO (for visualization purposes), but only 5 replicates are shown when apparently procedures were applied on bootstrapped data (n = 30 per group). If only 5 replicates were used for ANOSIM tests, this should be clearly stated.
  • In Line 165, please clarify if the body parts 1-6 were combined only within each individual seahorse. Did the combination head + trunk (body portion 1), for example, originally come from the same individual seahorse, in such a way that individuals were kept separate and identifiable throughout the statistical analysis? Or were the body parts of different individual seahorses mixed? This is central for the reader to identify how the sampling units were defined in the experimental design, as it is where a measure of the residual variation is estimated.

I strongly believe the study constitutes an important contribution of high value to the field and could have considerable effects on conservation programs for these and other ornamental marine species. If the issues on the statistical methods should be adequately clarified, the robustness of the findings would be strengthened, and the replicability of the investigation assured.

Author Response

Reviewer #2

The authors have carried out a long-time needed study to aide in the identification of the geographic origin of seahorses and/or their body parts. The manuscript provides a well-supported justification for its clearly stated objectives. The introduction is concise and complete and they have gone so far as to explicitly specify the questions that the study intends to answer. Results are discussed in a complete and organised way, the concluding remarks are distinct and in accordance with their main findings, and further topics to continue this investigation are properly recognised. However, I do have a few important concerns regarding the statistical analysis on the data and the way in which results are presented and interpretations obtained. Whilst the multivariate approach to analyse the data is perfectly justified, there are several points that in my opinion need to be clarified before the ms can be accepted.

We acknowledge Reviewer #2 for the constructive criticism and we have revised our manuscript to best accommodate the insightful suggestions provided. We hope that on its revised form Reviewer 2 considers that the issues highlighted have been successfully addressed.

Please note that all references to Line Numbers refer to the revised manuscript.

It is not clear why the authors applied univariate ANOVAs on the ratios of the elements, following the ANOSIM tests. The major advantage of a multivariate approach is the fact that it examines all variables (or descriptors) measured on n sampling units and their correlations in the same multidimensional space. This is, the multivariate procedure takes into account changes on all multidimensions at the same time. This, directly conflicts with univariate analyses, which take each variable in a single 1-dimensional space (i. e. as if all other variables did not exist) and are therefore bound to lose relevant information on the correlation between EF descriptors (which, in this case, are correlated because they were measured the same individuals). Results of ANOVAs are therefore not only redundant, but conflicting in their interpretation with the results of ANOSIMs and other multivariate procedures used (e.g. the multidimensional scaling used to visualize the data). (please refer to Chapter 1 in the Methods manual of PRIMER). If there is need to asses the relative contribution of EFs on the ability to distinguish the different groups, the authors should refer to methods such as SIMPER (used only in the case of section “3.2.1. Elemental composition of seahorse body portions”), or a close examination of the eigenvectors in the PCOs they have already applied.

As recommended by Reviewer 2, we have deleted all information and results related with the use of univariate ANOVAs on the ratios of the elements.

It is not clear why authors apply a series of sequential ANOSIM tests (Tables 1, 2 and 4) and interpreted their results as if all tests were truly independent (i.e. p values were not adjusted to correct for increased probability of type I error with a Bonferroni correction, for example). This procedure is not only unjustified, but its interpretation as a whole is unreliable from a probabilistic perspective. Moreover, hypotheses concerning differences between more than two groups are perfectly testable using ANOSIM, followed by pairwise tests between groups, especially if a strategy that reduces tests to a minimum is kept (please refer to Chapter 6 in the Methods manual of PRIMER).

We sincerely thank Reviewer 2 by the insightful comment that led the authors to revise and change their whole statistical analysis. We have decided to perform PERMANOVA instead of ANOSIM analysis. We have also clearly detailed what type of PERMANOVA (if one-way or two-way crossed) was used to test each specific question, followed by pair-wise comparisons. Please refer to subsection “2.4. Statistical analysis” that was fully re-written and that we hope is now clearer (and certainly more accurate from a statistical point of view).

PCOs are lacking the vectors that represent the descriptors, a visual aide that would be very informative on the partial contribution of each descriptor to the ordination map (i. e. which EFs contribute most to separate samples in the first and second principal coordinates) (please see Chapter 3 in the Methods manual of PERMANOVA)

As suggested we have now added the vectors to the PCOs (Figure 2 and Figure 3).

It is difficult to identify the different experimental hypotheses that are being tested (differences between locations, between body parts, between body fractions and locations, etc.), with each one of the different tests applied, and the particular fraction of data set used for such purposes. As a result, it is not clear whether the underlying design for ANOSIM tests is a one-way (1 factor), a 2-factor nested (1 factor nested in another), a 2-way crossed (2 factors combined with or without replication). The differences amongst these have important implications in terms of how the test are constructed and the permutations performed (please see Chapter 6 in the Methods manual of PRIMER).

Reviewer 2 is totally right! We have re-written our whole subsection “2.4. Statistical analysis”. Please note that we now use PERMANOVA and no longer ANOSIM. It now reads as follows (Lines 177-189): “To assess if the head, trunk, dorsal fin or tail were a reliable proxy of seahorses’ whole body, a one-way permutational analysis of variance (PERMANOVA) test [20] was per-formed, using body parts as the single factor (n = 5). A two-way crossed PERMANOVA was performed to evaluate the existence of significant differences in the EF of sea-horses using body parts as fixed factor with four levels (head, trunk, dorsal fin and tail) and sampling locations also as fixed factor with two levels (IMM and CCMAR) (n = 5). This same analysis was also performed to detect significant differences in the EF of seahorse body portions (specimens missing one or two body parts), belonging to sea-horses from each of the two locations. The PERMANOVA tests were run with 9999 Monte Carlo permutations. The pseudo-F values in the main tests and the t-statistic in the pairwise comparisons were evaluated in terms of the significance among levels of the tested factor. Values of p ≤ 0.05 reveal that the groups differ significantly.”.

There are some apparent contradictions in the number of replicates used for certain tests. For example, there is ANOSIM and a PCO (for visualization purposes), but only 5 replicates are shown when apparently procedures were applied on bootstrapped data (n = 30 per group). If only 5 replicates were used for ANOSIM tests, this should be clearly stated.

Reviewer 2 is right. On our revised version we have decided to no longer perform bootstrap resampling, so all analysis have n = 5 replicates. We have also replaced our CAP by a Random Forest analysis. The n of each analysis is now clearly indicated on subsection 2.4. Statistical analysis, as well as on the caption of each figure or table referring to the statistical analysis.

In Line 165, please clarify if the body parts 1-6 were combined only within each individual seahorse. Did the combination head + trunk (body portion 1), for example, originally come from the same individual seahorse, in such a way that individuals were kept separate and identifiable throughout the statistical analysis? Or were the body parts of different individual seahorses mixed? This is central for the reader to identify how the sampling units were defined in the experimental design, as it is where a measure of the residual variation is estimated.

To clarify this issue pointed by Reviewer 2 we have added the following text to the subsection 2.4. Statistical analysis (Lines 173-176): “All combinations of body portions were performed using portions from the same individual seahorse, thus allowing to always maintain individual specimens identifiable throughout the statistical analysis.”

I strongly believe the study constitutes an important contribution of high value to the field and could have considerable effects on conservation programs for these and other ornamental marine species. If the issues on the statistical methods should be adequately clarified, the robustness of the findings would be strengthened, and the replicability of the investigation assured.

We thank Reviewer 2 for the supporting words and hope that on its revised form the present study is now suitable for publication in Animals. Without the insightful comments provided we had not been able to revise our statistical analysis how it needed to be revised, nor had we clarified our hypothesis. This is what peer-review should always be like: providing guidance to improve submitted manuscripts and finding ways to support acceptance rather than rejection. Great review work by Reviewer 2!

Reviewer 3 Report

The paper "Successful use of geochemical tools to trace the geographic origin of long-snouted seahorse Hippocampus guttulatus raised in captivity" is an interesting and relevant study that is within the scope of the current study, in my opinion. The main problem is that it seems to be excessively long with some repeats and can be shortened in my opinion to become more reader-friendly.

Additionally, the measures to avoid contamination and ensure the quality were not described properly in the material and methods section. In line with that, the brands/grades of chemicals and standards used for sample preparation are not disclosed, which is important for at least potential reproducibility.

Another major comment: the authors refer to the supporting information which seems to be missing from the submission.

The authors state "All specimens were provided frozen (died from natural causes in the laboratory)...". At the same time, I don't see any ethical statement at the disclosure section.

Minor comments:

The paper seems to be original submission but contains a lot of track changes.

Lines 52-55 The sentence is hard to read, split it, please.

Line 56 The sentence is unclear.

Line 68 "termed as Data Deficient, which adds-up to the challenge of pursuing their conservation" is unclear.

Line 95 Quotation mark symbol seems to be redundant.

Line 132 What kind of silicate?

Line 157 What was the vendor of the standard?

Line 160 Present the average recoveries at least.

Line 322 Do you mean 'habitat' here?

Line 266 The sentence "In the specific case..." is unclear.

Author Response

Reviewer 3

The paper "Successful use of geochemical tools to trace the geographic origin of long-snouted seahorse Hippocampus guttulatus raised in captivity" is an interesting and relevant study that is within the scope of the current study, in my opinion. The main problem is that it seems to be excessively long with some repeats and can be shortened in my opinion to become more reader-friendly.

We thank Reviewer 3 for the constructive criticism and we hope that on its revised form we have been able to best accommodate the insightful suggestions provided.

Please note that all references to Line Numbers refer to the revised manuscript.

Additionally, the measures to avoid contamination and ensure the quality were not described properly in the material and methods section. In line with that, the brands/grades of chemicals and standards used for sample preparation are not disclosed, which is important for at least potential reproducibility.

All this relevant information was added to the revised article (please also see our reply to minor comments).

Another major comment: the authors refer to the supporting information which seems to be missing from the submission.

Reviewer 3 is right. There was some kind of error at the time of uploading this file and it is indeed missing. The authors apologize for this mistake that has impaired Reviewers to refer to our supplementary information. A single file with all supplementary information is now provided with the revised version of our manuscript.

The authors state "All specimens were provided frozen (died from natural causes in the laboratory)...". At the same time, I don't see any ethical statement at the disclosure section.

Reviewer 3 is right, the authors had missed that statement. We have now added the following section after “Funding” at the end of our manuscript (Lines 468-469): “Institutional Review Board Statement: Ethical review and approval were waived for this study as all specimens used were provided frozen and died from natural causes in the laboratory.”

Minor comments:

The paper seems to be original submission but contains a lot of track changes.

There was a lack of communication between the Editor and selected Reviewers informing them that the present manuscript was already a revised version of manuscript submitted to Animals. The Editor decided to consider this revised version as a new submission (an editorial decision that goes beyond the authors), hence the visible tack changes on the word file provided to Reviewers. We hope we have been able to clarify this issue raised by Reviewer 3 (any additional information may be retrieve from the Editor).

Lines 52-55 The sentence is hard to read, split it, please.

To best accommodate the suggestion by Reviewer 3 the sentence now reads as follows (Lines 53-55): “The fish skeletal system is formed by bones and cartilage; vertebrae are mainly com-posed by calcium, phosphate and carbonate and, at a smaller extent, by magnesium, sodium, strontium, lead, citrate, fluoride, hydroxide and sulfate [5].”.

Line 56 The sentence is unclear.

We have corrected the sentence and it now reads as follows (Lines 56-57): “Seahorses are flagship species and important ambassadors of marine conservation [6].”

Line 68 "termed as Data Deficient, which adds-up to the challenge of pursuing their conservation" is unclear.

For clarification and easier reading, we have deleted the last part of the sentence and it now reads as follows (Lines 66-69): “Managing seahorse fisheries and trade in a sustainable manner has proven to be a difficult task over the last decades, with several seahorse species still being termed as Data Deficient [9].”

Line 95 Quotation mark symbol seems to be redundant.

Quotation marks were deleted.

Line 132 What kind of silicate?

The sentence was corrected and now reads as follows (Lines 130-132): “The mortar grinder and the mortar and pestle were cleaned between samples with quartz powder followed by alcohol (70%), to avoid potential cross-contamination [19].”

Line 157 What was the vendor of the standard?

The requested information was added to the revised text and it now reads as follows (Lines 157-158): “Elements were quantified through calibrations with standard solutions of each analyte (High-Purity Standards - ISC Science).”

Line 160 Present the average recoveries at least.

We thank Reviewer 3 for noticing that this important information was missing. Average recovery rates were added to the revised version of our work (Lines 160-163) and it now reads as follows: “Accuracy of the ICP-MS method was evaluated by the analysis of a certified reference material BCS-CRM-513 (SGT Limestone 1). Recovery of elements was acceptable: Mg (116%), Ca (120%), Mn (118%), Fe (93%), Sr (119%), Ba (103%).”.

Line 322 Do you mean 'habitat' here?

Exactly, correctly as suggested.

Line 266 The sentence "In the specific case..." is unclear.

We think Reviewer 3 was referring to Line 366. The sentence has been revised (Lines 362-363) and for clarity it now reads as follows: “. In fact, none of the dorsal fins from seahorses cultured at IIM were correctly assigned to their place of origin (Table 3)..”

Round 2

Reviewer 2 Report

The authors have adequately responded to all my comments and questions regarding the statistical analysis. All points have been satisfactorily clarified so that methods can be now replicated and robust biological theory constructed. There is only a minor (really minor) suggestion to use the name "two-factor analysis of variance" or "full factorial 2-way analysis of variance" to refer to the corresponding analyses in the manuscript. The name "crossed" is usually associated with two-way designs that do not asses the interaction terms, this is, that assume the variation contributed by the interaction is negligible. Results presented herein prove this was not the case. Whilst this may seem a mere formality, I believe the consistent use of technical language is clarifying and helpful for all readers.

Reviewer 3 Report

I thank the authors for addressing my comments

This manuscript is a resubmission of an earlier submission. The following is a list of the peer review reports and author responses from that submission.

Round 1

Reviewer 1 Report

Please use "bony" instead of bonny (multiple times at least in lines 47, 82, 159, 298, 359, 378, 382)

How did you quantify the elements not represented in the CRM like Ba, Cu, Ni, Na, K?

You have 5 individuals from which you got the different parts, and 14 variables - I suggest first select the relevant parameters and re-run the analytical work, as 14 variables for 10 samples seem a bit overdetermining the result.

Moreover it is not clear, why and how the relevant parameters from literature Sr, Ba (not certified), Mg, Mn are different in the bony structures and why - why did you not measure the water...? But also meabye the food to determine the source of variation.#

I would like to see a much clearer description of the variation of the different parameters than overdoing statistics with 5 samples per site. Statistics can add, but first of all one needs a clear representation and discussion of the variation of the elemental concentrations in the different parts. 

Are water data available?

How can the differences explained? No water data are available. That is a major flaw of the experiment, when trying to establish the geographical origin. Which elements are expected to vary in the natural environment, and myabe compared to farms?

The differences in element /Ca ratios need to be better discussed.

I would like to see the statistics reduced to the most singificant parameters first, and consider the isue of overdetermining the classification by too many variables. What are the most important ones? Which ones have been found to vary in nature mainly (you say Ba, Sr, Mg, Mn) - why not discuss the variation of these in more detail?

Author Response

Dear Reviewer #1

We would like to acknowledge you constructive criticism that helped us improve our manuscript.

Attached please find our point-by-point reply to all reviewers comments.

We hope that on its revised form our work is now suitable for publication on Animals.

Reviewer 2 Report

In this paper the authors attempt to use the chemical composition of the body parts of sea horses to identify their geographic origin. While I found the paper interesting, there are three issues that I feel need to be addressed.

First, the authors did not discuss how variable the chemical composition of seawater is at a broad scale. In fact, they did not describe the chemical composition of the water in which their samples were reared. In order to assess the value of this technique, the authors need to establish that there are detectable differences in water chemistry across a geographic range that is relevant for seahorse conservation and that those water chemistry differences are reflected in the body composition of seahorses.

Second, there should be at least some discussion regarding the amount of variation within locations as compared to the observed differences between locations. If the differences among individuals collected from the same location are substantial, the accuracy of assignment may be limited. Similarly, it would be helpful to know if the same fish was assigned to the wrong location based on the analysis of different body parts (i.e. are there some fish that are being misassigned or are some body parts that are unreliable?). Variation among individuals within sampling locations may also explain why fish from one site were much easier to assign than from the other.

Finally, I think the authors should outline how this approach might be applied broadly in a conservation context. Would individuals of known origin need to be collected and analyzed to establish a baseline or could that be done by collecting a water sample? At what resolution might assignments be accurate and what level of resolution is required to protect imperiled stocks?

Minor editorial suggestions

Bony and bone throughout not bonny and bonne

L49 – Seahorses’ bodies are…

L92 – sentence is awkwardly worded: …EF may allow researchers to confirm…

L145 & 146 – HNO3  3 is not subscripted

L174 – I would suggest adding a sentence to explain that an R of 1 indicates dissimilarity

L369 – Seahorses usually

Table 1 formatting is odd, headers are out of place

Author Response

Dear Reviewer #2

We would like to acknowledge you constructive criticism that helped us improve our manuscript.

Attached please find our point-by-point reply to all reviewers comments.

We hope that on its revised form our work is now suitable for publication on Animals.
